# Mode-locked short pulses from an 8 μm wavelength semiconductor laser

Johannes Hillbrand [1,2✉], Nikola Opačak [1], Marco Piccardo [2,3], Harald Schneider [4], Gottfried Strasser [1], Federico Capasso [2] & Benedikt Schwarz [1,2✉]

Quantum cascade lasers (QCL) have revolutionized the generation of mid-infrared light. Yet, the ultrafast carrier transport in mid-infrared QCLs has so far constituted a seemingly insurmountable obstacle for the formation of ultrashort light pulses. Here, we demonstrate that careful quantum design of the gain medium and control over the intermode beat synchronization enable transform-limited picosecond pulses from QCL frequency combs. Both an interferometric radio-frequency technique and second-order autocorrelation shed light on the pulse dynamics and confirm that mode-locked operation is achieved from threshold to rollover current. Furthermore, we show that both anti-phase and in-phase synchronized states exist in QCLs. Being electrically pumped and compact, mode-locked QCLs pave the way towards monolithically integrated non-linear photonics in the molecular fingerprint region beyond 6 μm wavelength.

[1] Institute of Solid State Electronics, TU Wien, Guß, Vienna, Austria. [2] Harvard John A. Paulson School of Engineering and Applied Sciences, Harvard University, Cambridge, MA, USA. [3] CNST - Fondazione Istituto Italiano di Tecnologia, Via Pascoli 70/3, 20133 Milano, Italy. [4] Institute of Ion Beam Physics and Materials Research, Helmholtz-Zentrum Dresden-Rossendorf, Dresden, Germany. ✉email: johannes.hillbrand@tuwien.ac.at; benedikt.schwarz@tuwien.ac.at

The discovery of ultrashort light pulses has led to numerous breakthroughs in science and technology, including frequency combs[1], high-speed optical telecommunication[2] and refractive surgery in ophthalmology[3]. Nowadays, optical pulses are routinely generated in mode-locked lasers operating in the visible or near-infrared range[4,5]. Currently, large efforts are aimed at bringing ultrafast laser science in the mid-infrared (MIR) region to a similarly high degree of maturity[6]. Due to the lack of suitable gain media, methods for the generation of pulses in the molecular fingerprint region beyond 5 μm wavelength have so far relied on non-linear downconversion of near-infrared pulses[7]. Established techniques such as optical parametric oscillators[8] or difference frequency generation[9,10] either require sophisticated optical setups with tabletop dimensions or are restricted to mW-level of output power.

Quantum cascade lasers[11] (QCL) have matured to become the dominant MIR laser source. While being microchip-sized and electrically pumped, they are capable of producing Watt-level average power[12,13]. Quantum engineering of the active region allows to tailor the emission wavelength throughout the entire mid-infrared region. Hence, harnessing high-performance QCL technology for the generation of MIR pulses represents a long-sought milestone in ultrafast laser science. Mode-locked QCLs could serve as monolithic pump lasers for microresonators and resonant supercontinuum generation[14], paving the way towards broadband and high-brightness frequency combs. So far, the sub-picosecond carrier transport in the active region of MIR QCLs has constituted a seemingly insurmountable obstacle for the formation of short light pulses[15–17]. Both the upper-state lifetime and the gain recovery time of MIR QCLs are on the picosecond to sub-picosecond timescale[18], which is much shorter than the cavity roundtrip time. Thus, MIR QCLs favor a quasi-continuous intensity waveform rather than short pulses[19]. Note that in THz QCLs, where gain recovery times between 5 ps and 50 ps have been reported[20], actively mode-locked pulses were observed in several works[21–24] with durations as short as 4 ps[25].

To date, the only successful attempt of mode-locking in monolithic MIR QCLs was observed using a specially designed active region with strongly enhanced upper-state lifetime of the lasing transition[17]. However, the necessary design modifications limited mode-locked operation to cryogenic temperatures and peak power below 10 mW, thus impeding their practical use. Recently, active mode-locking of a QCL in an external ring cavity was reported[26]. This approach allows to mitigate the detrimental effect of spatial-hole-burning and enables a large modulation depth by modulating the entire QCL instead of just a short section. While mode-locked operation was observed at room temperature, the average power was limited to 3 mW and the pulse duration was more than 70 ps, with an estimated peak power below 0.5 W.

In this work, we demonstrate the generation of picosecond pulses in high-performance and monolithic 8 μm wavelength QCLs at room temperature both experimentally and theoretically. Mode-locking is achieved by electrically modulating the intracavity loss[27,28] using a short modulation section designed for efficient radio-frequency (RF) injection (Fig. 1a). Two different interferometric techniques are used to confirm that mode-locked operation is achieved over the entire lasing range. Furthermore, we show that both anti-phase and in-phase synchronization states can be excited by by varying the modulation frequency. Finally, a high-speed photodetector monolithically integrated on the laser chip allows to measure the first three harmonics of the laser beatnote and shows that their amplitude is enhanced by almost two orders of magnitude in the mode-locked regime.

## Results

**Mode-locking and synchronization states in bi-functional QCLs.** In order to achieve the large modulation depth required for stable mode-locking, close attention has to be paid to the band structure of the QCL active region. This effect is illustrated in Fig. 1a, b. As the bias applied to a standard QCL structure is decreased, it does not switch to absorption at the lasing wavelength, but becomes nearly transparent for the intracavity light due to a bias-dependent shift of the electronic levels, known as Stark effect (Fig. 1b). Hence, the modulation depth is severely limited in standard QCL designs. For this reason, we employ a bi-functional active region whose lasing wavelength and absorption wavelength at zero bias were matched to each other[12,29] (Fig. 1c). This strategy allows to overcome the aforementioned limitations of the modulation depth caused by the Stark shift. Most importantly, the bi-functional design shows excellent overall performance, which is competitive with other state-of-the-art designs. A 3.5 mm long device mounted epitaxial-side up emits more than 130 mW average power in continuous wave at room temperature (Fig. 2d).

As a first step towards pulse generation, it is essential to determine the optimal modulation frequency. For this purpose, mode-locking can be seen as synchronization of coupled oscillators[30] (Fig. 1e). Each pair of neighboring cavity modes creates a beating at their difference frequency, which is equal to the cavity roundtrip frequency. These beatings can be seen as oscillators coupled by the non-linearity of the gain medium. Thanks to this coupling, the cavity modes of a free-running QCL can be locked together without modulation, thus giving rise to a self-starting frequency comb[19]. Yet, this kind of frequency comb does not emit isolated pulses, but rather a quasicontinuous wave accompanied by a strong linear frequency chirp[31]. This corresponds to anti-phase synchronization and will be called frequency modulated (FM) comb in the following. It should be mentioned that another approach for generating pulses from QCL FM combs is to compensate their linear chirp using an external grating compressor[32]. However, this approach has to overcome the challenge that the chirp of the QCL is not perfectly linear, which results in relatively long pulses. In contrast, in-phase synchronization of the intermode beatings leads to the formation of short pulses. It was observed in several other lasers that the beating frequencies of the in-phase and anti-phase states are different[30,33]. As a consequence, while the cavity roundtrip frequency of the FM QCL comb $f_{\mathrm{rep}}^0$ may seem like a reasonable choice, we expect the optimal modulation frequency for generating pulses to differ from $f_{\mathrm{rep}}^0$.

In order to investigate these two synchronization states experimentally, we start by operating the laser well above its threshold current. Subsequently, the DC bias of the modulation section is decreased to 2.8 V, where the large absorption caused by the bifunctional design (Fig. 1c) brings the QCL just slightly below lasing threshold. In these conditions, modulation at the right frequencies can provide enough additional gain to reach threshold. Figure 1f shows the laser power depending on modulation frequency and power. At 33 dBm modulation power, the QCL reaches threshold when modulating close to $f_{\mathrm{rep}}^0$. Strikingly, a second modulation frequency where lasing occurs is observed almost 60 MHz higher than $f_{\mathrm{rep}}^0$, as predicted by the picture of synchronized oscillators. However, the theoretical reason for the splitting between the in-phase and anti-phase synchronization frequencies is not yet fully understood. Both the range around the two synchronization frequencies, where lasing is observed, as well as the optical power grow upon increasing the modulation power.

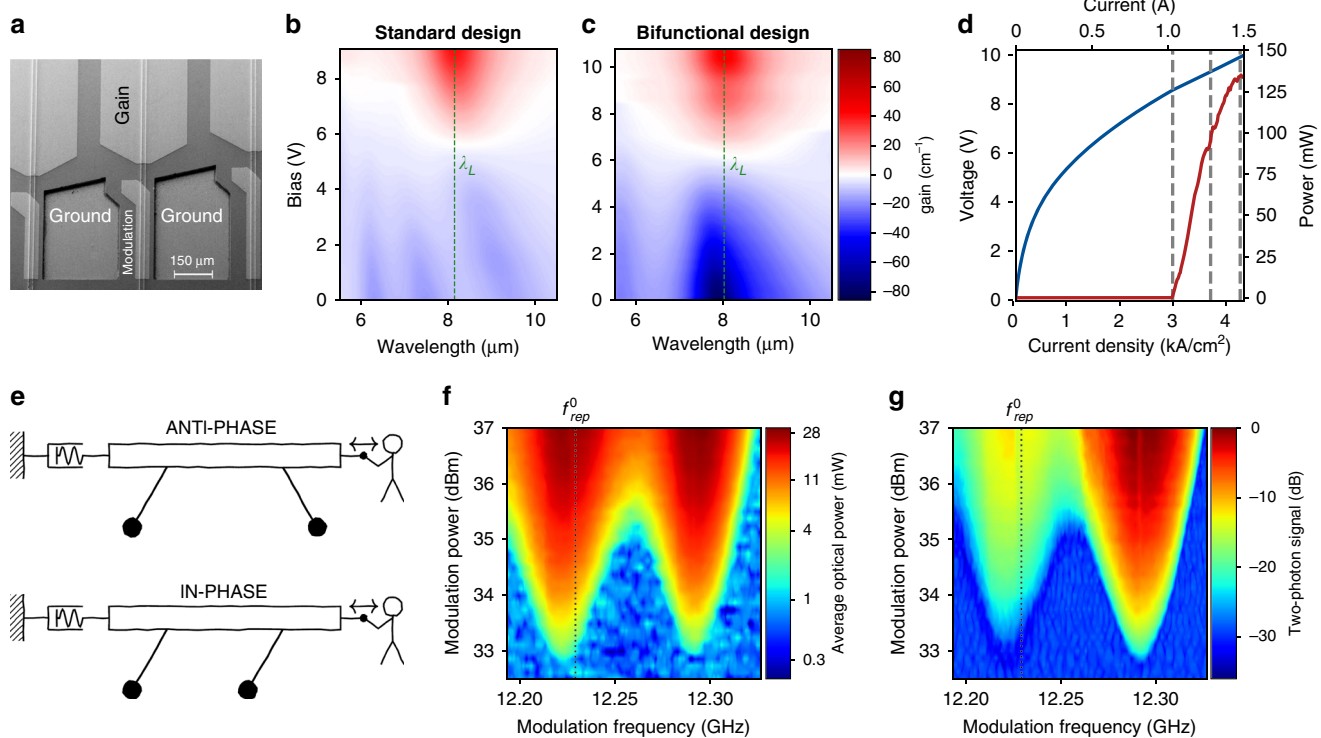

**Fig. 1 Bi-functional quantum cascade lasers for mode-locking. a** Scanning electron microscope image of three adjacent laser ridges. Each laser consists of a roughly 3 mm long gain section and a shorter (320–480 μm) modulation section. **b** Simulated gain and loss spectrum in a standard active region design[46] depending on the applied bias. Upon decreasing the bias, the structure becomes almost transparent at the lasing wavelength $\lambda_L$, limiting the maximally achievable modulation depth. **c** Simulated gain and loss spectrum in a bi-functional active region design[12], allowing to tune the gain at 10 V continuously to absorption (shown as negative gain) at 0 V. **d** Measured light–current–voltage (L–I–V) characteristics of an epi-up mounted bi-functional QCL at 15 °C. **e** Illustration of a system of coupled oscillators. This system shows an in-phase and anti-phase synchronization state, which oscillate at different frequencies depending on the coupling. Without external stimulation, the anti-phase state is more favorable due to the damped coupling. However, both synchronization frequencies can be probed by exerting mechanical force on the platform coupling the oscillators. In the QCL, the oscillators are represented by the intermode beatings, which tend to synchronize in anti-phase due to gain damping[19,30]. Both synchronization frequencies are probed by applying modulation to the laser. **f** Average optical power depending on the modulation frequency and power. Two synchronization states at $f_{rep}^0$ and 60 MHz above are observed. **g** Signal of a 2-QWIP sensitive to peak power as function of modulation frequency and power. The strongly increased signal of the lobe at $f_{rep}^0 + 60$ MHz indicates in-phase synchronization.

Even more insight is provided by using a two-photon quantum well infrared photodetector (2-QWIP) to detect the emitted light. The signal of the 2-QWIP is proportional to the square of the intensity. This allows to identify which modulation frequency leads to in-phase and which to anti-phase synchronization (Fig. 1g). Again, two lobes appear around $f_{rep}^0$ and 60 MHz above. Yet, the 2-QWIP signal is more than an order of magnitude larger in the lobe at higher $f_{mod}$. At this frequency, the laser operates in the in-phase synchronization regime and emits intense pulses, which leads to a strongly increased 2-QWIP signal.

**Interferometric pulse characterization.** In order to unequivocally prove mode-locking, we employ two independent methods to characterize the pulse dynamics at three points of operation from threshold up to the rollover current. Firstly, an interferometric RF technique called "Shifted wave interference Fourier transform spectroscopy" (SWIFTS)[34] is used to measure the phases of the QCL spectrum (details in Methods section). This information not only enables the reconstruction of the temporal waveform, but also allows to assess the phase-coherence of the pulses and whether they form a frequency comb. Secondly, we measure the interferometric autocorrelation (IAC) of the pulses using the 2-QWIP, which constitutes an additional well-

established proof for mode-locking and the pulse width. Fig. 2a shows the SWIFTS characterization of the QCL operated close to threshold. In contrast to the free-running laser, the intensity spectrum consists of a single Gaussian-shaped lobe. The SWIFTS spectrum represents the part of the intensity spectrum which is beating exactly at the modulation frequency. Since the SWIFTS spectrum has the same shape as the intensity spectrum over its entire span, the QCL generates a frequency comb whose repetition frequency is given by the modulation frequency. The intermodal difference phases $\Delta\phi$, which correspond to the spectral group delay, are synchronized almost perfectly in-phase. Hence, all parts of the spectrum have the same group delay and form a pulse. Indeed, the reconstructed waveform in Fig. 2b shows the emission of 6.5 ps short pulses. The full-width-half-maximum (FWHM) of the reconstructed pulses is given by the transform limit of the spectrum in Fig. 2b, indicating that there is negligible chirp in the pulses. In these optimal conditions, the peak power reaches almost 250 mW, which constitutes an enhancement of more than 12 compared to the average power of 20 mW. It is noteworthy that the pulse width does not appear to be limited by the gain bandwidth. Away from the optimal condition, also broader spectra could be obtained, however, these coincide with broader not transformation limited pulses. (see Supplementary Fig. 3).

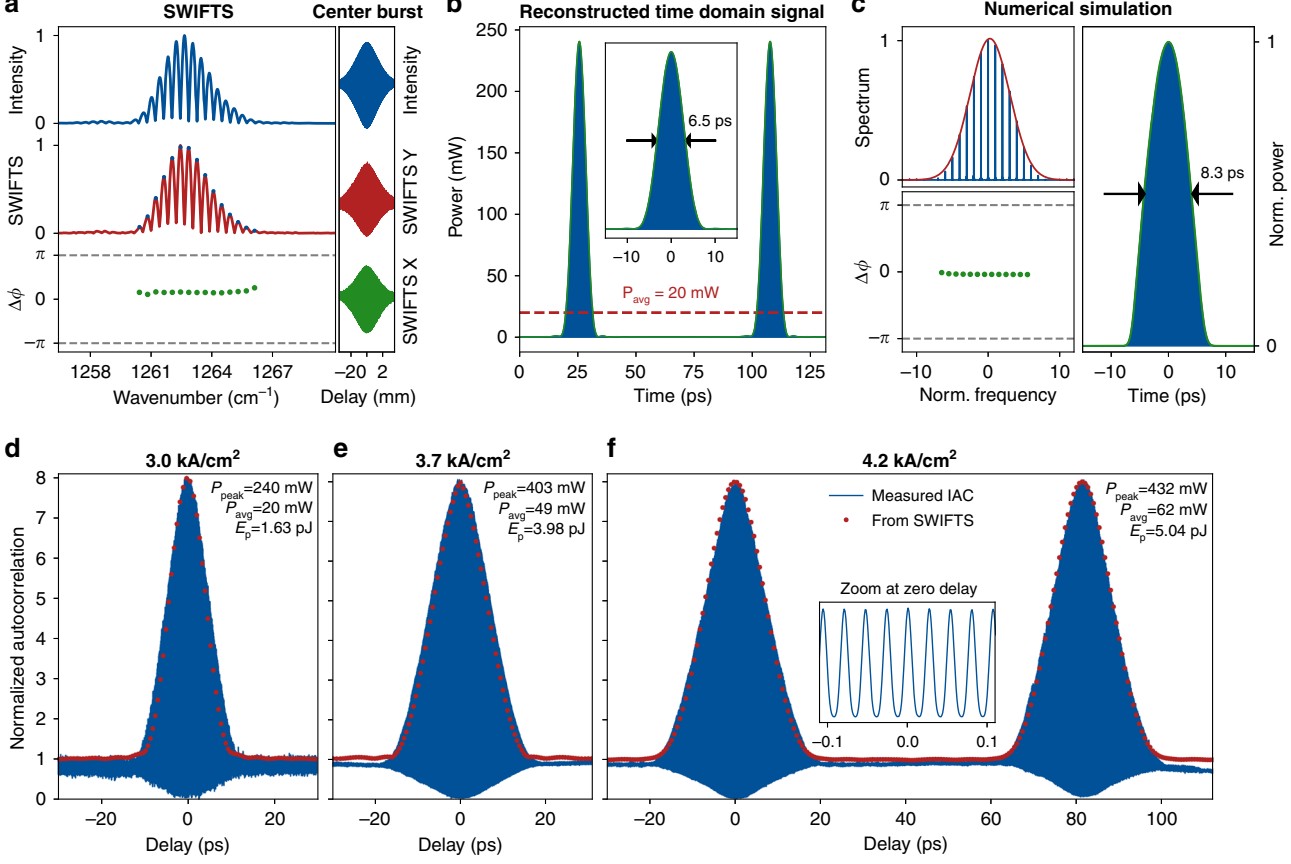

**Fig. 2 Mode-locked pulses from an 8 μm wavelength QCL. a** SWIFTS characterization of the QCL operated close to lasing threshold. The laser is modulated at the in-phase synchronization frequency and at 37 dBm power level. **b** Reconstructed time-domain signal of the QCL showing a train of 6.5 ps short pulses, which is 2.5 ps longer than the shortest pulses generated in THz QCLs[25]. **c** Simulation of the QCL using the coherent Master equation described in Supplementary Section 1. **d** Interferometric autocorrelation (IAC) of the QCL pulses close to threshold. Red dots: envelope of the IAC reconstructed using SWIFTS. **e** IAC at higher current. **f** IAC at the rollover current, still displaying the 8:1 ratio. The second burst at a delay equal to the cavity roundtrip time is due to the interference of subsequently emitted pulses. Its peak value of 8 provides another proof for the coherence of the pulses because phase-decoherence would smear out the fringes of the IAC and thus decrease the peak value to smaller than 8. Inset: zoom on the interferometric fringes. The DC bias applied to the modulation section was between 3 and 4 V for all measurements shown in this figure.

In order to model the cavity dynamics, we use a fully coherent master equation[35] (Supplementary Section 1). This single equation for the complex field replaces the entire Maxwell-Bloch system[36,37] and reliably predicts the spectral shape, phase relationship and pulse width observed experimentally (Fig. 2c). Furthermore, it allows experimentally unavailable analyses, e.g. the influence of dispersion and nonlinearities (Supplementary Figs. 2, 3, and 7).

The IAC close to threshold (Fig. 2d) shows a prominent peak at zero path difference caused by constructive interference of the pulses after the Michelson interferometer. The ratio of this peak to the background at a delay larger than the pulse width is 8:1, which is generally regarded as the smoking gun of mode-locked pulses. Encouragingly, the measured IAC is in excellent agreement with the expected IAC, which was calculated using the pulses obtained by SWIFTS (red dots in Fig. 2d). This confirms successful mode-locking and the retrieved pulse width[38].

The generation of pulses becomes increasingly challenging at higher gain current. Due to gain saturation, the wings of a pulse experience more gain than the peak. This effect leads to pulse broadening and can destabilize mode-locking. Fortunately, the large modulation depth provided by the bi-functional quantum design enables mode-locking over the entire lasing range from threshold to rollover. The IAC traces at 3.7 kA/cm² and at the

rollover current still show the required peak-to-background ratio of 8:1. The pulses at rollover are slightly broadened to roughly 12 ps, which is attributed partially to a slight chirp and partially to the gain saturation effect mentioned above (Supplementary Fig. 6). Yet, the average power is greatly increased to 62 mW, which results in over 430 mW peak power and 5 pJ pulse energy —more than an order of magnitude higher than recent reports of comparable mid-infrared semiconductor lasers emitting at shorter wavelengths[33,39].

**On-chip detector for beatnote characterization**. Another fascinating aspect of bi-functional quantum design is the possibility to monolithically integrate ultrafast photodetectors. While this is particularly important in applications such as photonic integrated circuits, it also provides a tool to measure the beatnote with very large signal-to-noise ratio directly on the chip (Fig. 3a). This provides crucial information about the type of synchronization state and about its stability. To this end, we have fabricated 7 mm long devices divided into three sections: the modulation and gain sections are used for generating the pulses and a third section close to the opposite facet of the modulation section is used as on-chip detector. The length of this detection section was reduced to 70 μm to minimize its parasitic capacitance, which increases the electrical detection bandwidth. RF probes are landed directly on

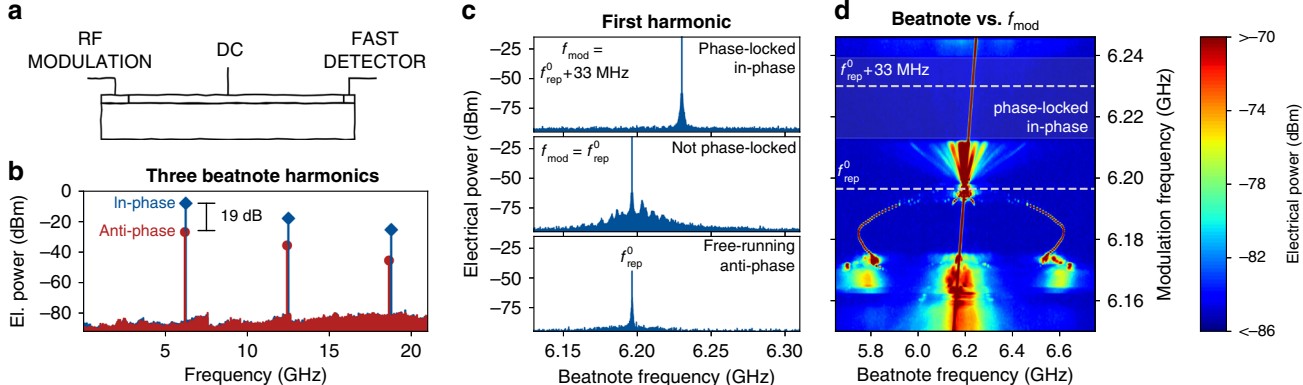

**Fig. 3 Synchronization under strong modulation. a** Schematic of a 3-section QCL comprised of modulation, gain and high-speed detector sections. **b** First three harmonics of the beatnote of the free-running 7 mm long QCL FM comb (red) compared to the actively mode-locked QCL (AM comb, blue) in logarithmic scale. **c** Laser beatnote while free-running (bottom), at $f_{mod} = f_{rep}^0$ (middle) and at $f_{mod} = f_{rep}^0 + 33$ MHz (top). While a broad pedestal is visible for $f_{mod} = f_{rep}^0$, the beatnote is perfectly locked for $f_{mod} = f_{rep}^0 + 33$ MHz. **d** RF spectrum around $f_{rep}^0 = 6.196$ GHz as the modulation frequency is varied around $f_{rep}^0$. The phase-noise of the RF spectrum disappears abruptly at $f_{mod} \approx f_{rep}^0 + 20$ MHz, corresponding to in-phase synchronization. Here, the beatnote consists of a single narrow peak, indicating that the laser is phase-locked to the modulation. Furthermore, the sharp sidepeaks visible at $f_{mod} = 6.18$ GHz are attributed to a periodic modulation of the QCL output, as previously observed in simulations[16].

the detection section to extract the laser beatnote. Figure 3b shows the first three harmonics of the beatnote in the free-running and the actively mode-locked regime. In the latter conditions, the beatnote amplitudes increase by 19 dB due to the much larger amplitude modulation. The residual beatnote in the free-running FM regime can be attributed to the amplitude-phase coupling in the active medium[35]. The zoom on the first harmonic beatnote (Fig. 3c) allows to assess the phase-coherence and stability of the frequency comb. The free-running QCL is operating in the anti-phase state showing a weak beatnote at $f_{rep}^0$. Previous work[40–42] has shown that a weak electrical modulation can be used to lock and stabilize the beatnote of the anti-phase state. However, the situation is very different when applying strong modulation at $f_{rep}^0$. In this case, the modulation enforces an AM waveform, which is contrary to the natural anti-phase behaviour of the laser. As a result, the beatnote of this waveform is not phase-locked, as indicated by the pedestal around $f_{mod}$. The situation changes completely, when the modulation frequency is tuned to the synchronization frequency of the in-phase state ($f_{rep}^0 + 33$ MHz). There, the strong modulation is in consensus with the natural behavior of the laser, leading to a phase-locked frequency comb with narrow beatnote. Figure 3d shows the beatnote map, while continuously sweeping the modulation frequency across the in-phase and anti-phase synchronization frequencies. The upper two subfigures of Fig. 3c correspond to slices taken at these two frequencies, which is indicated by the dashed lines. The frequency of the beatnote is controlled by the modulation over the entire span shown in Fig. 3d. However, a phase-locked comb is only generated around the in-phase synchronization frequency, highlighted by the shaded gry area. There, the beatnote consists of a single narrow line locked to the modulation frequency, indicating the formation of a fully coherent comb. In contrast, when modulating at $f_{rep}^0$, a noise pedestal is visible around the modulation frequency.

## Discussion
Our experiments provide unambiguous proof for the generation of mode-locked pulses in high-performance QCLs at room temperature - a goal which remained elusive since the invention of the QCL—and confirm stunning similarities to synchronization

in coupled oscillators. To our knowledge, these mode-locked QCLs constitute the first compact and electrically pumped source for ultrashort pulses beyond 5 μm wavelength, demonstrating that they are a highly promising technology for ultrafast laser science in the long-wave infrared region. The availability of such a source paves the way towards a semiconductor-based platform for non-linear photonics, potentially enabling broadband mid-infrared frequency combs and supercontinuum generation.

The next steps could involve the inclusion of a third section that provides an additional knob to alter the phase and dispersion in the cavity. Initial numerical simulations already predict a pulse width reduction of more than a factor two using this approach. A possible alternative direction can be the combination with an external pulse compressor[32]. In this case, the laser could be operated in a regime where a residual but perfectly linear chirp is accepted in order to optimize for a larger spectral span and more efficient use of the gain. Possible all-solid state solutions could utilize nonlinear effects to produce solitons or solitary structures by pumping a Kerr microresonator[43] or directly by utilizing hybrid mode-locking of the recently reported phase turbulence ring QCLs[44,45].

## Methods
**QCLs optimized for RF modulation.** The QCLs are processed as buried heterostructures. The width of the QCL ridges is 12 μm and the facets are left as cleaved. The area of the top contact of the modulation section is minimized to decrease its parasitic capacitance, which increases the RF injection efficiency. Ground contacts for the modulation are provided by etching through the Fe-doped InP layer next to the laser ridges down to the grounded n⁺-InP substrate. These ground contacts are required for efficient RF injection via coplanar tips. The modulation signal is provided by a HP8341B synthesized sweeper and amplified up to roughly 5 W. The insertion loss at 12 GHz is 15 dB including cables and bias-tee.

**SWIFTS and IAC.** The light emitted by the QCL is shone through a Bruker Vertex 70v FTIR spectrometer. In order to perform SWIFTS, the light is then detected by a home-built fast QWIP at the exit of the FTIR. The optical beating obtained from the QWIP is subsequently amplified and mixed down to roughly 10 MHz using a local oscillator. A Zurich Instruments HF2LI lock-in amplifier and the trigger of the FTIR are used to record the SWIFTS and intensity interferograms in rapid scan mode. The IAC is obtained by detecting the pulses at the exit of the FTIR using the 2-QWIP and recording its photocurrent depending on the path delay of the FTIR. The peak power of the pulses is determined by normalizing the integral of the reconstructed SWIFTS waveform to the average power measured by a calibrated thermal power meter.

## Data availability

The data that support the plots within this paper and other findings of this study are available from the corresponding authors upon reasonable request.

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

## Acknowledgements

This work was supported by the Austrian Science Fund (FWF) in the framework of Building Solids for Function (Project W1243), the projects NanoPlas (P28914) and NextLite (F4909). This project has received funding from the European Research Council (ERC) under the European Union's Horizon 2020 research and innovation programme (Grant agreement No. 853014).

## Author contributions

J.H. processed the QCLs and carried out the experiments. B.S. and J.H. built up the SWIFTS and IAC setups. N.O. performed the simulations using the tool developed by N.O and B.S. M.P. carried out the temporal reconstruction using the IAC data. H.S. provided the 2-QWIP. J.H. wrote the manuscript with editorial input from N.O., B.S., G.S., and F.C. B.S. supervised the work. All authors contributed to analysing the results and commented on the paper.

## Competing interests

The authors declare no competing interests.
