## [Peer Review File · Nature Communications]

REVIEWER COMMENTS

Reviewer #1 (Remarks to the Author):

The manuscript reports about ps pulse generation in a QC laser using a technique which is already known in the literature of diode laser as "hybrid mode-locking". It is a way to combine passive and active mode-locking in a two- or multi-section device that was proposed at the beginning of the 90's in the group of Larry Coldren. Several many others reports have been produced since then, I believe that a reference citing one of these works would be appropriate. Similar experiments on two-section QC lasers were also published few years ago by the Capasso's group (ref. 17 in the manuscript) and showed mode-locking operation at 77K with pulse duration of 3 ps.

The results presented in this letter show mode-locking operation of a QC laser operating at room temperature. The data are of excellent quality and their analysis is convincing. It illustrates that a train of pulses from a QC laser can be obtained by modulating the device at a frequency which is not that of the natural beating between longitudinal modes. Indeed, in this case the modes are in counter phase therefore susceptible to generate phase rather than amplitude modulation. The laser has to be driven on a different beating frequency which enforces a constant phase, thus bringing them "in-phase", as the authors like to say. This qualitative interpretation is well justified by a theoretical modelling based on Maxwell-Bloch equations and third order nonlinearities, which has been recently published by the same authors.

The evidence of mode-locking operation is well proven and illustrated by comparing two independent techniques, SWIFTS and interferometric autocorrelation, which are in excellent agreement on the pulse duration of $\sim 7 - 12$ ps. The ensemble of the data presented shows also the resistance of the QC laser system to enter into the mode-locking operation. The external modulation has to be so large to "enforce" the laser into an operation "contrary to the natural anti-phase behaviour of the laser". About 5W of microwave power is injected into the small section to achieve mode-locking and it seems that with less than 2 watts nothing occurs. For a better evaluation of this point authors should give the current injected into the laser (not the current density). This would allow the reader to compare the cw-power with the RF-power necessary for mode-locking operation. What is clear already is that the laser should be turn on and off to get pulse generation. Given the amount of power injected it is not obvious that bi-functional active regions are really necessary.

The resilience of QC laser to enter into the pulse regime is also evident from the pulse duration, ~ 10 ps. Even, in the best situation only a very small amount of the longitudinal modes is brought "in phase". A pulse duration of 10 ps correspond to a bandwidth of ~ 100 μ eV, which is roughly 1% only of the available gain bandwidth. Talking about transform limited pulses is misleading. What it is said, is simply that one gets in the time space a duration equal to the number of modes that he has been able to lock in the frequency domain. Why should not be like this? The problem is the 99% of the band that has not been exploited.

In conclusion I believe that these results are interesting and can be published after the following revisions.

1) The laser characteristics have to be better presented:

- VI in Volt(A) and Watt(A), that will make clear the electrical power injected.
- A spectrum of the laser in cw and under 37dbm of modulation

2) A physical discussion that explains why 5 W of microwave power is necessary to achieve modelocking

3) Clarify why only few modes can be locked and use the term "transform limited" in an understandable way in order to avoid confusion between the locked longitudinal modes and the available gain bandwidth.

4) Fig. 3. The relation between the panels c) and d) it is not obvious. Why there is no signal in panel d) at $f_{\text{mod}} + 33$ MHz? In general, this figure needs to be better commented and results have to be explained in clearer way. This also to valorise the data and the technique used which are quite unique and related to the bi-functional active region.

Reviewer #2 (Remarks to the Author):

The manuscript reports the first room-temperature actively mode-locked mid-IR QCL operation. Despite an earlier demonstration of actively mode-locked mid-IR QCL at cryogenic temperatures (Ref. 17), this work represents significant improvement in the device design and performance in terms of operating temperature and power and it further provides a lot of new details on the actively-mode-locked QCL operation. Given novelty and high results quality as well as a strong interest to the subject of mid-IR frequency comb generation by both scientific and applications communities, I believe this paper should be accepted to Nature Communications.

The manuscript is generally very well written and I only have a few minor comments:

1. It is highly desirable to include laser spectra for a range of current densities shown in Fig. 2d.
2. The authors should explain in the main text why numerical simulations predict 30% broader laser pulses than what is observed experimentally. This is particularly puzzling as both SWIFT measurements and theory show nearly perfectly phase-locked pulses and nearly identical spectral bandwidth.
3. The authors should provide more details on the device configuration. In particular, the manuscript provides no details on the integrated detector implementation. The rationale for using side-ground plane for the modulation section rather than relying on grounding the entire device substrate is not explained. Furthermore, details of the laser structure (cladding layers thickness and doping, active region thickness, substrate doping, etc) are not provided.
4. There may be one reference missing in right column, line 9 on p. 4

Reviewer #3 (Remarks to the Author):

Hillbrand and authors report on modelocking of QCLs for the generation of mid-infrared pulses. Compared to any previous work, the authors use of bi-functional design that shows enhanced absorption at low applied fields to increase the modulation depth between gain and loss in the system. By applying a strong microwave modulation close to the repetition frequency, the authors show an enhanced peak power from the modelocked device. The authors show the clear generation of pulses, with considerably increased peak powers over the entire operating current. This is done with two complementary methods, SWIFTS that is recent, and IAC with a two photon QWIP, with a very nice agreement between them. Pulse widths of 6ps are shown. Finally an on-chip geometry with an integrated detector (using the characteristics of bi-functional design) show the beatnotes from the device, and the behaviour between i) free-running, ii) modelocked at f_{rep} and iii) modelocked close to f_{rep} showing how the device changes from anti-phase to in-phase synchronisation of the modes when shifting from free-running to active modulation.

I believe the work is of high quality with its main impact being the enhancement of the peak powers when modelocked with the use of the bi-directional design, and the use of two interferometric techniques to show and confirm pulse generation. However, I have strong reservations on the novelty of the results and the claims of ultrashort pulses. An important work by D. G. Revin et al, Nat. Comm 7, 11440 (2016) is not presented at all. Here, in an external geometry, shows active modelocking over the entire operational current of the QCL at room temperature similar to the claims here. This is also with a 'standard' QCL. There is also work by M. Singleton et al, Journal of the Optical Society of America B 36, 6, 1676 (2019) that needs to be compared with as this shows enhanced peak powers $\sim 600\text{mW}$ without active modulation. Further, there is a body of work that has been performed on QCLs operating in the THz. This includes amongst others work by:

1. S. Barbieri et al, Nat. Photonics 5, 306–313 (2011).

2. A. Mottaghizadeh et al, *Optica* 4, 168 (2017).
3. J. R. Freeman et al, *Opt. Express* 21, 16162 (2013).
4. F. Wang et al, *Laser Photonics Rev.* 11, 1700013 (2017).
5. F. Wang et al, *Optica* 2, 944 (2015).

Although there is a distinction between MIR and THz QCLs, the results are relevant as all these have shown active modelocking with short pulse generation, despite the short gain recovery time of these lasers that is shorter than the photon round trip time. They have also shown enhanced power with the active modulation, although I think not to the same degree as the current paper and not at room temperature. Further, the shortest pulses from active modelocked THz QCLs are 4ps (Wang et al (2017)). This is shorter than the current paper despite a much lower frequency i.e. ~ 10 oscillations of the electric field compared to over 200 for the current paper. Thus I believe the claim of ultrashort pulses here is not correct and should be corrected throughout.

I believe the paper could be enhanced by concentrating more on some interesting phenomena that appears in the data, especially the transition between the anti-phase state and the in-phase state. In the supplementary material, there are suggestions that the two synchronization states are already evident in the beatnote of the free-running laser and separated by 57 MHz in the RF spectrum. This correlates with the work in the main manuscript showing, that for pulse generation, the active modulation must be slightly detuned from the resonance frequency (that has been shown for THz QCLs). However, I did not find any in-depth discussion on this in the main paper, why this state that has both in- and anti- phase components and the reasons for the detuning. There is some text on this being well known but it was not clear for me. This could be clarified, with some material from the supplementary section brought into the main paper. The description in the supplementary section is quite brief and the figures discussed rapidly so I did not follow all the text.

Other points

- 1) It would be useful to have the SWIFT temporal (and spectral) scans for the free running QCL and for modulated at f_{rep} , to compare to figure 2b.
- 2) On a related issue, figure 3f (labelled 'd') of the supplementary shows a considerably different spectrum compared to figure 2a of the main manuscript. Were they taken at different currents? And if this is the case is there a reason the different response different
- 3) Some relevant references of other approaches to simulate the response of QCLs using Maxwell Bloch formalism could be added (e.g. P. Tzenov et al, *Opt. Express* 24, 23232 (2016), J. Freeman et al, *Phys. Rev. A* 87, 063817 (2013)). Further Z. Han et al, *Opt. Express* 28, 6002 (2020) has compared SWIFT to autocorrelation and is possibly relevant here.
- 4) The authors are driving the small section of such that the QCL is brought beyond and below threshold to generate pulses. This allows very strong modulation close to threshold resulting in the shortest pulses. Does this limit the pulse width that can be attained, just before the rollover point in figure 2f? Would more RF power be advantageous, or would there be a limit? How does this compare with active modelocking in interband lasers?
- 5) Regarding figure 3 on the three section device, a strong beatnote is (-30dBm) observed, suggesting some in-phase behaviour. Anti-phase behaviour should not generate such a strong fundamental beatnote if I have understood correctly. Could the authors comment on this? Further does this triple section device effect the beatnote? How does it compare to a single section QCL?
- 6) Further in figure 3, in the in-phase component shows an enhancement by 19 dB. However the device is being modulated very strongly (33dBm). Therefore, I would expect a contribution to the detected QCL beatnote from the RF modulation through for example GHz pick-up in the cables. Could the authors comment on this? Could the authors show the RF spectrum with the modulation when the QCL is just at threshold i.e. when there is no QCL beatnote?
- 7) The three beatnotes of fig 3 seem to have a different RF ratio between in-phase and anti-phase compared. Does this give any extra information on the coupled oscillator image?
- 8) The authors should state how the peak powers are determined.

To conclude, I think the paper should strongly consider the work in relation to work already done on MIR and

THz QCLs with an indepth discussion on the relevance and the differences with the current work. The paper should tone down the claim of 'ultra'-short pulse and if possible to develop in more depth the anti-phase to in-phase transition, and why a free running QCL possibly shows both. How to get to really ultrashort pulses and very high peak powers could be realistically mentioned in the conclusion and when dispersion needs to be considered. I would re-consider the paper after these changes.

Vienna, 31.7.2020

Regarding: Revised manuscript NCOMMS-20-09590

Dear reviewers,

We are grateful for your valuable comments and your fast response. Our manuscript was revised in order to address the comments of the reviewers. The changes are given point by point in the following (reviewer comments in bold, answers in normal font, manuscript text in italic).

Reviewer #1 (Remarks to the Author):

The manuscript reports about ps pulse generation in a QC laser using a technique which is already known in the literature of diode laser as “hybrid mode-locking”. It is a way to combine passive and active mode-locking in a two- or multi-section device that was proposed at the begging of the 90’s in the group of Larry Coldren. Several many others reports have been produced since then, I believe that a reference citing one of these works would be appropriate.

We thank the reviewer for making us aware of this work and have added the following two citations to the manuscript:

[1] Ebeling, K., Coldren, L., Miller, B. & Rentschler, J., ‘Generation of single-longitudinal-mode subnanosecond light pulses by high-speed current modulation of monolithic two-section semiconductor lasers’. Electronics Letters 18, 901 (1982).

[2] Bowers, J., Morton, P., Mar, A. & Corzine, S. ‘Actively modelocked semiconductor lasers’. IEEE Journal of Quantum Electronics 25, 1426-1439 (1989).

About 5W of microwave power is injected into the small section to achieve mode-locking and it seems that with less than 2 watts nothing occurs. For a better evaluation of this point authors should give the current injected into the laser (not the current density). This would allow the reader to compare the cw-power with the RF-power necessary for mode-locking operation. What is clear already is that the laser should be turn on and off to get pulse generation. Given the amount of power injected it is not obvious that bi-functional active regions are really necessary.

We agree with the reviewer that showing the absolute current injected into the laser will add additional information to the paper and allows the comparison between the RF and DC power in the modulation section. For this reason, we have added a second x-axis on top of the L-I-V curve showing the absolute current injected into the laser. We have left the second x-axis with the current density, so that readers can compare the threshold current density to other works.

Figure 1d of the main text

Furthermore, we have added a section to the supplementary material, where the RF insertion loss into a 300 μm short modulation is characterized using a microwave rectification technique. This allows experts to gain more insights into the internal mechanisms:

“Ensuring a high-speed frequency response of the modulation section is essential to achieve the large modulation depth required for active mode-locking. There are at least two parameters, which can influence the modulation capabilities of semiconductor lasers: the speed, up to which the gain medium can be modulated due to its internal carrier dynamics, and the capacitance of the laser. Due to fast intersubband scattering on the sub-picosecond timescale, QCLs can be modulated efficiently up to tens of GHz and do not show relaxation oscillations [9], in contrast to many interband lasers. Hence, the limiting property of the modulation response of QCLs is in most cases their parasitic capacitance.

We have measured the frequency response of a 300 μm short modulation section using a microwave rectification technique (supp. Fig. 8). At the cavity roundtrip frequency of a 3.5 mm long device of approximately 12.3 GHz, the modulation setup (including coaxial cables and RF tips used to inject the signal as well as the parasitic capacitance of the QCL) shows a loss of roughly 15 dB. Hence, around 160 mW of RF power are injected into the laser when the power of the modulation source is 5 W. This could be improved considerably by further minimizing the parasitic capacitance of the modulation section and using longer devices with a smaller cavity roundtrip frequency.”

Figure 2: Frequency response of a 300 μm short modulation section measured using microwave rectification

From this detailed analysis we can obtain the power injected into the laser after the RF insertion loss to be around 150 mW. For characterizing the QCL pulses shown in Fig. 2 of the main text, the DC bias of the modulation section is set to between 3-4 V, because we observe that the shortest pulses are generated under these conditions. The DC power applied to the modulation section is below 100 mW under these conditions, which is similar to the injected RF power.

Regardless of how efficient the laser can be modulated in terms its RF properties, it is also important that a modulation of the bias leads to a modulation of the round-trip gain. This is why we included figures 1b&c in the original manuscript, where we compare the bias dependency of the gain and absorption of a bi-functional with a non-bifunctional active region.. The bi-functional active region is specifically optimized such that the dominant optical transition matches the laser wavelength for the entire bias range. In such a device, a bias modulation leads to a large modulation of the round-trip gain at any bias point, also below the transparency current. In conventional QCLs, however, the optical transition tunes strongly with the applied bias and tunes out of resonance at lower bias. Thus, conventional QCLs cannot be efficiently modulated below their transparency current. However, we found that active mode-locking is clearly working the best when the short section is operated at lower bias e.g. 3-4V, which is below the transparency current. Hence, the regime, in which we observe the shortest pulses, would not be accessible with a non-bifunctional design. Particularly, when the long gain section is biased close to roll over, a low bias of the short section is a requirement to obtain isolated pulses. To clearly state the used operating conditions, we have added the following short statement to the figure caption of Fig. 2 of the main text:

“The DC bias applied to the modulation section was between 3-4 V for all measurements shown in this figure.”

The resilience of QC laser to enter into the pulse regime is also evident from the pulse duration, ~10 ps. Even, in the best situation only a very small amount of the longitudinal modes is brought “in phase”. A pulse duration of 10 ps correspond to a bandwidth of ~100 μ eV, which is roughly 1% only of the available gain bandwidth. Talking about transform limited pulses is misleading. What it is said, is simply that one gets in the time space a duration equal to the number of modes that he has been able to lock in the frequency domain. Why should not be like this? The problem is the 99% of the band that has not been exploited.

We agree with the reviewer that one of the current challenges is to further reduce the pulse width. Optimizing for a larger spectral span is a possible direction, however, we found that for this particular device different operating conditions that show a larger spectral span actually lead to a broadening of the pulse (e.g. see Suppl. Fig. 3). As the reviewer noted, this indeed shows that the gain bandwidth does not appear to be the limiting factor.

We added a new paragraph to the conclusion to discuss possible future directions on how to further optimize the device to get shorter pulses and larger spectra.

To our knowledge, the definition of transform-limited pulses is that their duration is limited by the bandwidth of the spectrum and not the chirp, which is consistent with the reviewers commented. We have added the following text for clarification:

“The full-width-half-maximum (FWHM) of the reconstructed pulses is given by the transform limit of the spectrum in Fig.2b, indicating that there is negligible chirp in the pulses. In these optimal conditions, the peak power reaches almost 250 mW, which constitutes an enhancement of more than 12 compared to the average power of 20 mW. It is noteworthy that the pulse width does not appear to be limited by the gain bandwidth. Away from the optimal condition, also broader spectra could be obtained, however, these coincide with broader not transformation limited pulses. (see suppl. Fig. 3).”

In conclusion I believe that these results are interesting and can be published after the following revisions.

1. The laser characteristics have to be better presented: -VI in Volt(A) and Watt(A), that will make clear the electrical power injected. -A spectrum of the laser in cw and under 37dbm of modulation

We thank the reviewer for this constructive comment and believe that the paper was improved considerably by adding the absolute current to Fig. 1d as well as the supplementary section shown above.

2. A physical discussion that explains why 5 W of microwave power is necessary to achieve mode-locking

We think that the supplementary section showing the RF characterization of the modulation section provides a detailed answer to this comment. The necessary RF power could be decreased considerably by further optimizing the insertion loss of the modulation section.

3. Clarify why only few modes can be locked and use the term “transform limited” in an understandable way in order to avoid confusion between the locked longitudinal modes and the available gain bandwidth.

We thank the reviewer for this valuable comment. In our opinion, the clarifying paragraph mentioned above improved the readability of the paper.

4. Fig. 3. The relation between the panels c) and d) it is not obvious. Why there is no signal in panel d) at $f_{\text{mod}} + 33$ MHz? In general, this figure needs to be better commented and results have to be explained in clearer way. This also to valorise the data and the technique used which are quite unique and related to the bi-functional active region.

We thank the reviewer for this comment. We are aware that fig 3d contains plenty of new exciting information, which leads to a certain complexity. We thus included fig 3c to highlight the most relevant parts – the difference between modulating at f_{rep} and $f_{\text{rep}}+33\text{MHz}$ (which corresponds to the in-phase resonance frequency).

The upper two subfigures of figure 3c represent slices of figure 3d taken at f_{mod} and $f_{\text{mod}}+33\text{MHz}$, which is indicated by the dashed lines.

In the phase locking regime, the signal is not directly visible as the beatnote is completely phase locked to the reference and thus exactly matching the injection frequency. The fact that not even a pedestal is visible clearly shows the excellent purity of the locked state.

In order to better describe the key points we rewrote the corresponding paragraph to:

“Fig. 3d shows the beatnote map, while continuously sweeping the modulation frequency across the in-phase and anti-phase synchronization frequencies. The upper two subfigures of fig. 3c correspond to slices taken at these two frequencies, which is indicated by the dashed lines. The frequency of the beatnote is controlled by the modulation over the entire span shown in fig. 3d. However, a phase-locked comb is only generated around the in-phase synchronization frequency, highlighted by the shaded grey area. There, the beatnote consists of a single narrow line locked to the modulation frequency, indicating the formation of a fully coherent comb. In contrast, when modulating at f_{rep} , a noise pedestal is visible around the modulation frequency.”

Reviewer #2 (Remarks to the Author):

1. It is highly desirable to include laser spectra for a range of current densities shown in Fig. 2d.

Thank you for this comment. We show 4 different laser spectra for increasing current density of the DC section in the supplementary material (see figure below). Furthermore, the experimental results are compared to numerical simulations. Both the simulations and experimental data show that the pulses are broadened with increasing pumping current, which is caused by larger net gain window that the pulse experiences. At the same time, the phases remain synchronized almost perfectly in-phase. Hence, the pulse broadening visible in the interferometric autocorrelation traces in Fig. 2 of the main text is caused by a narrowing of the spectrum and not by the occurrence of a chirp. A chirp would appear when moving away from the optimal operating conditions.

Figure 3 simulated (left) and measured spectra (right) for increasing pumping current.

2. The authors should explain in the main text why numerical simulations predict 30% broader laser pulses than what is observed experimentally. This is particularly puzzling as both SWIFT measurements and theory show nearly perfectly phase-locked pulses and nearly identical spectral bandwidth.

We attribute the small deviations of the the pulse widths between simulation and experiment to the way how we include the RF modulation in model. In order to better explain this origin, we modified the corresponding paragraph in the supplementary material to:

“The current in the modulation section is modeled with a simple sinusoidal modulation

$$J = J_{DC} + J_{mod} \sin(\omega_{mod} t), \quad (2)$$

where ω_{mod} is the modulation frequency, J_{DC} represents the DC bias and the amplitude of the modulation depth is given with J_{mod} . The transparency current could be added directly. The small deviations of the pulse widths and shapes between simulation and experiment can be attributed to the linear dependence between the current and the sinusoidal modulation in equation (2). A more complex model that uses a bias dependent tunneling rate, similarly to Ref. [2] could further improve the quantitative agreement of the pulse shape and width.”

3. The authors should provide more details on the device configuration. In particular, the manuscript provides no details on the integrated detector implementation. The rationale for using side-ground plane for the modulation section rather than relying on grounding the entire device substrate is not explained. Furthermore, details of the laser structure (cladding layers thickness and doping, active region thickness, substrate doping, etc) are not provided.

Thank you for this constructive comment. Firstly, we have inserted a paragraph, which describes more details on the integrated detector implementation:

“To this end, we have fabricated 7 mm long devices divided into three sections: the modulation and gain sections are used for generating the pulses and a third section close to the opposite facet of the modulation section is used as on-chip detector. The length of this detection section was reduced to 70 μm to minimize its parasitic capacitance, which increases the electrical detection bandwidth. RF probes are landed directly on the detection section to extract the laser beatnote.”

Regarding the reason for the ground contacts on the side of the modulation top contact, we have reformulated the corresponding paragraph in the Methods section:

“Ground contacts for the modulation are provided by etching through the Fe-doped InP layer next to the laser ridges down to the grounded n+-InP substrate. These ground contacts are required for efficient RF injection via coplanar tips.”

Furthermore, we have added another section to the supplementary material, which describes the layer structure of the QCL in detail, including substrate, cladding, active region and contact layers. We are confident that these changes improved the readability of the manuscript and are grateful for this review comment.

4. There may be one reference missing in right column, line 9 on p. 4

We thank the reviewer for pointing this out and have corrected the mistake.

Reviewer #3 (Remarks to the Author):

I believe the work is of high quality with its main impact being the enhancement of the peak powers when modelocked with the use of the bi-directional design, and the use of two interferometric techniques to show and confirm pulse generation. However, I have strong reservations on the novelty of the results and the claims of ultrashort pulses. An important work by D. G. Revin et al, Nat. Comm 7, 11440 (2016) is not presented at all. Here, in an external geometry, shows active modelocking over the entire operational current of the QCL at room temperature similar to the claims here. This is also with a 'standard' QCL.

We are grateful for this comment and fully agree that the work of D. G. Revin et al. represents an important work and needs to be cited accordingly. The citation has been added to the manuscript along with the following paragraph:

"Recently, active mode-locking of a QCL in an external ring cavity was reported [18]. This approach allows to mitigate the detrimental effect of spatial-hole-burning and enables a large modulation depth by modulating the entire QCL instead of just a short section. While mode-locked operation was observed at room temperature, the average power was limited to 3 mW and the pulse duration was more than 70 ps."

However, we disagree with the reservations on the novelty of our results raised by the reviewer. While the work by D. G. Revin et al. certainly is important, both the system and the approach described in our manuscript are entirely different. Instead of an external cavity, we use a microchip-sized monolithic cavity. The advantages of our approach are evident, showing pulse durations more than 10 times shorter with up to 20 times more average power than in the work by D.G. Revin et al. Furthermore, we show for the first time the existence of two oscillation modes of the modulated QCL and that the optimal modulation frequency differs from the beatnote frequency of the free-running QCL. We prove that the bi-functional design provides sufficient modulation depth that mode-locking in a monolithic cavity becomes possible without modulating the entire chip, as it was done by Revin et al. We provide a detailed characterization using the two independent methods (SWIFTS and interferometric autocorrelation), which is generally regarded as the definite proof for mode-locking and has not been shown yet.

Finally, we want to emphasize that the senior author of the paper by D. G. Revin et al., A. Belyanin, confirms the novelty of our results by citing our manuscript in the following way:

"Active mode locking schemes are less convenient and produce pulses of limited peak power [5–10], although there are some recent advances in this direction: see [11], where a large modulation depth was achieved by careful design of the active region."

Y. Wang and A. Belyanin, arXiv:2005.12475 (2020)

There is also work by M. Singleton et al, Journal of the Optical Society of America B 36, 6, 1676 (2019) that needs to be compared with as this shows enhanced peak powers ~ 600mW without active modulation.

We thank the reviewer for this comment. Indeed, the approach chosen by M. Singleton in the group of J. Faist is another way to generate pulses from a free-running QCL comb emitting a quasi-continuous wave. The idea of this approach is to use a grating compressor to compensate the almost linear chirp of the QCL output. However, since the chirp is not perfectly linear, this approach results in relatively

long pulses (12 ps) despite the large spectral bandwidth of the QCL frequency comb, which limits the use of this technique. Furthermore, the optical losses in the grating compressor are so large, that the peak power of the generated pulses are not higher than the peak power of the incident light.

*“While the peak to average ratio increases to 40.7 at the optimal location, a 10-fold increase from the uncompressed output, **losses in the experiment mean that there is no overall increase in peak power.**”*

M. Singleton et al., JOSA B (2019)

Most importantly, the grating compressor setup is a table top setup which is not comparable to a monolithic QCL.

Nevertheless, we agree with the reviewer that this approach should be mentioned in the manuscript, because it could become very interesting provided that the problem of the non-linear chirp can be solved. We have therefore added the citation and a corresponding paragraph:

“It should be mentioned that another approach for generating pulses from QCL FM combs is to compensate their linear chirp using an external grating compressor. However, this approach has to overcome the challenge that the chirp of the QCL is not perfectly linear, which results in relatively long pulses.”

Further, there is a body of work that has been performed on QCLs operating in the THz. This includes amongst others work by:

- 1. S. Barbieri et al, Nat. Photonics 5, 306–313 (2011).**
- 2. A. Mottaghizadeh et al, Optica 4, 168 (2017).**
- 3. J. R. Freeman et al, Opt. Express 21, 16162 (2013).**
- 4. F. Wang et al, Laser Photonics Rev. 11, 1700013 (2017).**
- 5. F. Wang et al, Optica 2, 944 (2015).**

Although there is a distinction between MIR and THz QCLs, the results are relevant as all these have shown active modelocking with short pulse generation, despite the short gain recovery time of these lasers that is shorter than the photon round trip time. They have also shown enhanced power with the active modulation, although I think not to the same degree as the current paper and not at room temperature.

We thank the reviewer for pointing out that the original manuscript did not yet discuss the great advances that have been made in actively mode-locked THz QCLs. We have added all citations suggested by the reviewer to the revised manuscript along with the following paragraph:

“So far, the sub-picosecond carrier transport in the active region of MIR QCLs has constituted a seemingly insurmountable obstacle for the formation of short light pulses [15,16,17]. Both the upper-state lifetime and the gain recovery time of MIR QCLs are on the picosecond to sub-picosecond timescale [18], which is much shorter than the cavity roundtrip time. Thus, MIR QCLs favor a quasi-continuous intensity waveform rather than short pulses [19]. This situation is very different in THz QCLs, where the gain recovery time and upper-state lifetime are an order of magnitude longer [20]. As a consequence, actively mode-locked pulses from THz QCLs have been reported in several works [21,22,23,24] with durations as short as 4 ps [25].”

We believe that including previous work on actively mode-locked THz QCLs adds valuable information to the manuscript and also highlights the clear differences between THz and MIR QCLs in terms of laser dynamics.

Further, the shortest pulses from active modelocked THz QCLs are 4ps (Wang et al (2017)). This is shorter than the current paper despite a much lower frequency i.e. ~ 10 oscillations of the electric field compared to over 200 for the current paper. Thus I believe the claim of ultrashort pulses here is not correct and should be corrected throughout.

We agree that the work of Wang et al. demonstrates that dispersion compensation has great potential for generating very short pulses. Further work will have to be done to reach such short pulse durations with MIR QCLs. However, we do not agree that the claim of ultrashort pulses is incorrect. R. Paschotta defines this term in the following way:

“There is no commonly accepted definition of “ultrashort”, but usually this label applies to pulses if their pulse duration is at most a few tens of picoseconds, and often in the range of femtoseconds.”

R. Paschotta, Encyclopedia of Laser Physics, https://www.rp-photonics.com/ultrashort_pulses.html

Particularly for semiconductor lasers, we are confident that 6.5 ps short pulses can be called ultrashort. Nevertheless, we have added the following figure caption to highlight the large potential of mode-locked QCLs demonstrated in the THz region:

“Reconstructed time-domain signal of the QCL showing a train of 6.5 ps short pulses, which is 2.5 ps longer than the shortest pulses generated in THz QCLs [25].”

I believe the paper could be enhanced by concentrating more on some interesting phenomena that appears in the data, especially the transition between the anti-phase state and the in-phase state. In the supplementary material, there are suggestions that the two synchronization states are already evident in the beatnote of the free-running laser and separated by 57 MHz in the RF spectrum. This correlates with the work in the main manuscript showing, that for pulse generation, the active modulation must be slightly detuned from the resonance frequency (that has been shown for THz QCLs). However, I did not find any in-depth discussion on this in the main paper, why this state that has both in- and anti- phase components and the reasons for the detuning. There is some text on this being well known but it was not clear for me. This could be clarified, with some material from the supplementary section brought into the main paper. The description in the supplementary section is quite brief and the figures discussed rapidly so I did not follow all the text.

We agree with the reviewer that the splitting between the beatnote frequencies of the in-phase and anti-phase states is an intriguing phenomenon. As the reviewer mentions, similar phenomena have been observed in actively mode-locked interband cascade lasers and THz QCLs, as well as passively mode-locked quantum dot lasers. One of the key messages of our manuscript is that these two synchronization states also exist in QCLs and play an essential role in finding the right modulation frequency for active mode-locking. We are confident that this message is very well supported by the experimental evidence presented in figures 1 and 3 of the main text.

However, the theoretical reason for the splitting between the in-phase and anti-phase synchronization frequencies is not yet fully clear. We have attempted to model this phenomenon using the Maxwell-Bloch simulation tool. The simulations are in excellent agreement with nearly every aspect of the experimental observations, as can be seen in Figs. 2 and 3 of the supplementary material for example. While the Maxwell-Bloch simulation tool indeed predicts that the optimal modulation frequency should be slightly different from the beatnote frequency of the free-running comb, the splitting between the two synchronization frequencies is not fully clear yet.

We agree with the reviewer that the sentence “It is well known from coupled oscillators that the in-phase and anti-phase states synchronize at different frequencies depending on the coupling.” Of the original manuscript is an oversimplification of the observed phenomena. We have therefore changed it accordingly:

“It was observed in several other lasers that the beating frequencies of the in-phase and anti-phase states are different.”

Other points

1. It would be useful to have the SWIFT temporal (and spectral) scans for the free running QCL and for modulated at f_{rep} , to compare to figure 2b.

We have added this figure to the supplementary material, as suggested by the reviewer.

Figure 4 Comparison of the SWIFTS characterization and reconstructed waveform of the free-running and actively mode-locked QCL frequency comb. *a: the spectrum of the actively mode-locked QCL consists of a single Gaussian-shaped lobe. The intermodal difference phases are synchronized in-phase. b: reconstructed waveform of the actively mode-locked QCL, showing a train of short pulses. c: the spectrum of the free-running QCL consists of several lobes. In contrast to a, the intermodal difference phases are linearly played over the full range of 2π . d: reconstructed waveform emitted by the free-running QCL frequency comb showing almost no amplitude modulation.*

2. On a related issue, figure 3f (labelled 'd') of the supplementary shows a considerably different spectrum compared to figure 2a of the main manuscript. Were they taken at different currents? And if this is the case is there a reason the different response different

We apologize for the mistake and corrected the label of the figure. The shape of the spectrum depends strongly on the modulation frequency. When the modulation frequency is swept across the beatnote frequency of the free-running laser f_{rep} , a kind of anti-crossing of the spectrum occurs (as displayed in supplementary Fig. 2). When the modulation frequency is close to this anti-crossing, the spectrum shows sidelobes, exactly as the reviewer commented. Interestingly, this anticrossing is not only visible when modulating at the anti-phase frequency f_{rep} , but also and the in-phase frequency $f_{\text{rep}}+60$ MHz.

3. Some relevant references of other approaches to simulate the response of QCLs using Maxwell Bloch formulism could be added (e.g. P. Tzenov et al, Opt. Express 24, 23232 (2016), J. Freeman et al, Phys. Rev. A 87, 063817 (2013)). Further Z. Han et al, Opt. Express 28, 6002 (2020) has compared SWIFT to autocorrelation and is possibly relevant here.

We have added the citations according to the reviewer's recommendation.

4. The authors are driving the small section of such that the QCL is brought beyond and below threshold to generate pulses. This allows very strong modulation close to threshold resulting in the shortest pulses. Does this limit the pulse width that can be attained, just before the rollover point in figure 2f? Would more RF power be advantageous, or would there be a limit? How does this compare with active modelocking in interband lasers?

We agree with the reviewer that it will be essential to investigate the mechanisms currently limiting the pulse width in order to reach even shorter pulses. In fact, it is predicted by the classical active mode-locking theory of Kuizinga and Siegman that the pulse width grows as the gain increases. This is because the pulse broadening effect caused by gain saturation becomes stronger at higher gain current. At the same time, the theory of Kuizinga and Siegman predicts that the pulse width only depends very weakly on the modulation depth M , namely proportionally to $M^{-1/4}$. We also observe this experimentally. Above a certain power level of the injected modulation signal, both the pulse shape and with hardly change. Still, at this stage it is difficult to judge to which extent a larger RF power (or further optimization of the RF response) can be used to further improve the performance. If more RF power is available, a possible way could be to increase the length of the modulation section. However, we believe that adding an additional phase section would be the first step and this will be part of our follow-up work.

The main differences between QCLs and interband lasers are that interband lasers have a much larger upper state lifetime and are known to support passive mode-locking. In the picture of coupled oscillators, this means that the in-phase synchronization state can be naturally stable. Hence, in interband lasers it is also much easier access the in-phase state by active modulation. Furthermore, interband lasers generally show a strong absorption below transparency due to the interband transition and the sign of the stark shift. Hence, specific optimizations such as the bi-functional QCL active region are mostly not necessary.

QCLs, on the other hand, need a delicate engineering of both the bandstructure (bi-functional active region) and the RF coupling, because a much stronger modulation is required to actively lock the in-phase comb state. A main advantage thereby is the naturally much better RF modulation response of QCLs.

While we find the reviewers questions of point 4 very interesting, we are afraid that a detailed discussion on this topics is mostly appealing to experts of the community. We thus prefer to leave a detailed discussion to a follow up work together with a study of a several of different geometries and here focus on describing the main advances in an appealing way to a broad audience.

5. Regarding figure 3 on the three section device, a strong beatnote is (-30dBm) observed, suggesting some in-phase behaviour. Anti-phase behaviour should not generate such a strong fundamental beatnote if I have understood correctly. Could the authors comment on this? Further does this triple section device effect the beatnote? How does it compare to a single section QCL?

In principle a perfect anti-phase state would have a vanishing beatnote, similarly as in a perfect in-phase state the chirp would vanish. In real devices (and also in the numerical simulations) residual beatnotes or residual chirps are mostly observed. However, the main behavior is still described by either of the two regimes. Key is, which regime the laser is trying to optimize. From our theoretical work (Opacak et al PRL 2020) on QCL FM combs, we know that there always will be a small residual amplitude modulation due to the coupling of the phase to the amplitude. Due to this coupling the amplitude and phase modulations cannot be entirely decoupled. From numerical simulation we found that by dropping the responsible coupling term the residual amplitude modulation in the FM regime vanishes. However, this would correspond to an oversimplified solution, similarly to neglecting a residual chirp in the in-phase (AM) regime.

We added the following note to the reference to the corresponding paragraph:

“The residual beatnote in the free-running FM regime can be attributed to the amplitude-phase coupling in the active medium [opacak2019theory].”

6. Further in figure 3, in the in-phase component shows an enhancement by 19 dB. However the device is being modulated very strongly (33dBm). Therefore, I would expect a contribution to the detected QCL beatnote from the RF modulation through for example GHz pick-up in the cables. Could the authors comment on this? Could the authors show the RF spectrum with the modulation when the QCL is just at threshold i.e. when there is no QCL beatnote?

We thank the reviewer for this valuable comment. Indeed, some effort was needed to minimize the crosstalk. We have performed the experiment suggested by the reviewer and measured the signal extracted from the detector section when the QCL is just below threshold (corresponding to the RF crosstalk) and above threshold.

As this experiment shows, the pick-up signal is more than 20 dB weaker than the beatnote extracted from the detector section. We have added this figure to the supplementary material.

7. The three beatnotes of fig 3 seem to have a different RF ratio between in-phase and anti-phase compared. Does this give any extra information on the coupled oscillator image?

This is a very intriguing question. We agree that this information might be useful to enhance the oscillator picture. However, the first step would be to directly connect the oscillator picture to the Maxwell-Bloch relations, which is a main aim of our current research. This would directly show if this additional information can be used. Currently, we have the following picture to describe the beatnote amplitude behavior. In the anti-phase state the beatnote originates from the amplitude-phase coupling. Thus it is related to the amplitude response of the gain medium to the chirped FM modulation. In the in-phase regime the beatnotes are directly governed by the kind of synchronization. From an experimental point, it would be interesting to extract this information also from even higher harmonics, despite the known difficulties. However, this information should also be contained in the waveform obtained by SWIFTS.

8. The authors should state how the peak powers are determined.

The peak powers are determined by setting the integral of the reconstructed SWIFTS waveform equal to the average power measured using a thermal power meter. We have added the following sentence to clarify this:

“The peak power of the pulses is determined by normalizing the integral of the reconstructed SWIFTS waveform to the average power measured by a calibrated thermal power meter.”

To conclude, I think the paper should strongly consider the work in relation to work already done on MIR and THz QCLs with an indepth discussion on the relevance and the differences with the current work. The paper should tone down the claim of 'ultra'-short pulse and if possible to develop in more depth the anti-phase to in-phase transition, and why a free running QCL possibly shows both. How to get to really ultrashort pulses and very high peak

powers could be realistically mentioned in the conclusion and when dispersion needs to be considered.

We are confident that the additional paragraphs we have added to the revised manuscript provide a detailed list of the different results, which have been obtained with actively mode-locked THz QCLs. We believe that adding this information considerably improved the paper by highlighting the striking differences of THz and MIR QCLs in terms of laser dynamics. According to the reviewer's suggestion, we have added the following new paragraph to the conclusion:

“The next steps could involve the inclusion of a third section that provides an additional knob to alter the phase and dispersion in the cavity. Initial numerical simulations already predict a pulse width reduction of more than a factor two using this approach. A possible alternative direction can be the combination with an external pulse compressor [Singleton et al. Optica 2019]. In this case, the laser could be operated in a regime where a residual but perfectly linear chirp is accepted in order to optimize for a larger spectral span and more efficient use of the gain. Possible all-solid state solutions could utilize nonlinear effects to produce solitons or solitary structures by pumping a Kerr microresonator [Obrzud et al. Nature Photonics 2017] or directly by utilizing hybrid mode-locking of the recently reported phase turbulence ring QCLs [Piccardo et al. Nature 2020, Meng et al. Optica 2020].”

We hope that with this letter, the revised manuscript and the supplementary information we could address all concerns of the reviewers.

Sincerely,

Benedikt Schwarz

REVIEWER COMMENTS

Reviewer #1 (Remarks to the Author):

Second Review report on the manuscript by J. Hillbrand and co-workers

The authors have suitably answered to my questions and have clarified the text where some difficult interpretations were arising. I regret that they have not found the space for showing the spectrum of the free-running QCL, which is, as expected, substantially different than that under amplitude modulation. Indeed, this very interesting work not only demonstrates that the laser can be actively mode-locked, but also that the transition between two very distinct comb regimes is achievable: the natural FM situation, with a linear phase relation, and the actively AM where the phase is constant. They should not be afraid if one spectrum is 10 times wider than the other, what matters is their difference and the possibility of reaching both regimes.

In any case the paper is a very good work and can be published as it is.

Reviewer #2 (Remarks to the Author):

I am satisfied with the changes introduced in the revised manuscript and believe the manuscript can now be accepted for publication in Nature Communications.

Reviewer #3 (Remarks to the Author):

I thank the authors in responding extensively to my previous queries. The work is of high quality and opens up interesting observations and analysis of QCL operation.

I have the following remarks.

- 1) The authors state in the rebuttal that 'One of the key messages of our manuscript is that these two synchronization states also exist in QCLs'. As this is a key point, I suggest that it should therefore be mentioned in the abstract/conclusion i.e. "We show that both anti- and in- phase synchronized states exist in QCLs". I believe this is an important message, beyond short pulse generation.
- 2) My last question on these states - regarding the splitting in RF frequency between the synchronized states, could this simply be down to dispersion (as possibly shown in ref 24 and 25), where the RF modulation compensates for any dispersion, when no dispersion compensation is used?
- 3) I thank the authors in acknowledging the work of D.G. Revin et al. The authors should compare the peak power (hundreds of mW) and not the average power, as the cavity is considerably larger for work of Revin et al, and as the peak power is mentioned for reference 17 (i.e. use the same metric).
- 4) Regarding gain recovery times in THz QCLs, there are considerable lower values reported compared to that cited here (ref 20). See S. Markmann et al, Opt. Exp. 25, 21753 (2017), D. R. Bacon Appl. Phys. Lett. 108, 081104 (2016), F. Wang et al, Optica 2, 944 (2015) etc (from 5 ps to 18ps), longer than MIR QCLs, but not an order of magnitude greater. Further, a short gain recovery time does not prevent active modelocking. Indeed a short gain recovery time should be advantageous for active modelocking as it makes it 'easier' to modulate at high frequencies. Therefore the phrase 'This situation is very different in THz QCLs, where the gain recovery time and upper-state lifetime are an order of magnitude longer. As a consequence, actively mode-locked pulses from THz QCLs have been reported in several works [21,22,23,24] with durations as short as 4 ps [25].' is not strictly correct. The authors should correct this.
- 5) Although the term ultrashort can be used in a very wide context, I do not agree here on the use of 'ultrashort' here. The authors will surely achieve shorter pulses in the near future (below 1ps?), resulting in difficulty in distinguishing the future work with the current work. I suggest that the authors use 'short' instead of 'ultrashort' in the title.
- 6) The authors should state in the manuscript 'However, the theoretical reason for the splitting between the

in-phase and anti-phase synchronization frequencies is not yet fully clear.'

7) Regarding my query on figure 3, the authors show that there is an enhancement of 20dB of the beatnote above threshold compared to below threshold. This shows there is ~ 50 dB of RF pickup. This means that in figure 3b, the 'in-phase' beatnote is actually a lot smaller than the 'anti-phase' beatnote, if the RF pick up is taken into account. If this interpretation is correct, what are the implications on the data? Does this just mean that there are fewer modes that contribute to the RF signal for the in-phase state?

Regarding: Revised manuscript NCOMMS-20-09590

Dear reviewers,

We are grateful for your valuable comments and your fast response. Our manuscript was revised in order to address the comments of the reviewers. The changes are given point by point in the following (reviewer comments in bold, answers in normal font, manuscript text in italic).

Reviewer #3 (Remarks to the Author):

1) The authors state in the rebuttal that ‘One of the key messages of our manuscript is that these two synchronization states also exist in QCLs’. As this is a key point, I suggest that is should therefore be mentioned in the abstract/conclusion i.e. “We show that both anti- and in-phase synchronized states exist in QCLs”. I believe this is an important message, beyond short pulse generation.

We agree with the reviewer that this point should be emphasized and have therefore added the following sentence to the abstract:

“Furthermore, we show that both anti-phase and in-phase synchronized states exist in QCLs.”

2) My last question on these states - regarding the splitting in RF frequency between the synchronized states, could this simply be down to dispersion (as possibly shown in ref 24 and 25), where the RF modulation compensates for any dispersion, when no dispersion compensation is used?

We agree that dispersion might play a role in the observed splitting between the two synchronization states. However, we believe that also other parameters such as the non-linearity of the gain medium play an important role. For example, it is possible that the effective refractive index of the waveguide depends on the intensity resulting in different roundtrip times for the AM and FM states. However, as the reviewer pointed out in a comment below, the origin of this splitting is not yet fully clear from a theoretical point of view and more theoretical and experimental work will be necessary in the future.

3) I thank the authors in acknowledging the work of D.G. Revin et al. The authors should compare the peak power (hundreds of mW) and not the average power, as the cavity is considerably larger for work of Revin et al, and as the peak power is mentioned for reference 17 (i.e. use the same metric).

We agree with the reviewer that this information allows a better comparison of the work of Revin et al. with our results. While the results reported by Revin et al. do not include a reconstruction of the pulse shape, an upper limit for the peak power can be found from the following paragraph of their manuscript:

“The maximum out-coupled peak optical power is estimated to be 12 mW if the emission pulses are indeed 3–4 ns long. However, it is likely that this duration is limited by the resolution of the detector, since the same pulses are observed for all types of the emission spectra, both narrow and broad ones. The width of the observed narrow peak spectra under modulation below the round-trip frequency is 0.45 cm⁻¹, which should correspond to pulses of 75 ps duration and with a peak power of several hundred mW, assuming perfect mode locking.” Revin et al., Nat. Comm. 7 (2016)

We conclude that the upper limit of the peak power reported by Revin et al. can be estimated by $12 \text{ mW} \times 3 \text{ ns} / 75 \text{ ps}$, which equals 480 mW. As suggested by the reviewer, we have added this information to the manuscript:

“While mode-locked operation was observed at room temperature, the average power was limited to 3 mW and the pulse duration was more than 70 ps, with a maximum estimated peak power below 0.5 W.”

4) Regarding gain recovery times in THz QCLs, there is considerable lower values reported compared to that cited here (ref 20). See S. Markmann et al, Opt. Exp. 25, 21753 (2017), D. R. Bacon Appl. Phys. Lett. 108, 081104 (2016), F. Wang et al, Optica 2, 944 (2015) etc (from 5 ps to 18ps), longer than MIR QCLs, but not an order of magnitude greater. Further, a short gain recovery time does not prevent active modelocking. Indeed a short gain recovery time should be advantageous for active modelocking as it makes it ‘easier’ to modulate at high frequencies. Therefore the phrase ‘This situation is very different in THz QCLs, where the gain recovery time and upper-state lifetime are an order of magnitude longer. As a consequence, actively mode-locked pulses from THz QCLs have been reported in several works [21,22,23,24] with durations as short as 4 ps [25].’ is not strictly correct. The authors should correct this.

We agree with the reviewer that there are several works reporting slightly different values for the gain recovery time and upper-state lifetime in THz QCLs. We have therefore modified the sentence mentioned by the reviewer:

“This situation is very different in THz QCLs, where the gain recovery time and upper-state lifetime are longer than in MIR QCLs”

5) Although the term ultrashort can be used in a very wide context, I do not agree here on the use of ‘ultrashort’ here. The authors will surely achieve shorter pulses in the near future (below 1ps?), resulting in difficulty in distinguishing the future work with the current work. I suggest that the authors use ‘short’ instead of ‘ultrashort’ in the title.

We thank the reviewer for this comment and agree that in future work it should certainly be possible to decrease the pulse duration. We have therefore modified the title of the manuscript accordingly:

“Mode-locked short pulses from an 8 μm wavelength semiconductor laser”

6) The authors should state in the manuscript ‘However, the theoretical reason for the splitting between the in-phase and anti-phase synchronization frequencies is not yet fully clear.’

We agree with the reviewer and have added this sentence to the manuscript:

“However, the theoretical reason for the splitting between the in-phase and anti-phase synchronization frequencies is not yet fully understood.”

7) Regarding my query on figure 3, the authors show that there is an enhancement of 20dB of the beatnote above threshold compared to below threshold. This shows there is ~ 50dB of RF pickup. This means that in figure 3b, the ‘in-phase’ beatnote is actually a lot smaller than the ‘anti-phase’ beatnote, if the RF pick up is taken into account. If this interpretation is correct, what are the implications on the data? Does this just mean that there are fewer modes that contribute to the RF signal for the in-phase state?

We want to point out that all beatnotes are plotted on a logarithmic scale. Hence, the power of the RF crosstalk below threshold is 20 dB or a factor of 100 weaker than the beatnote extracted from the detector section above threshold. Therefore, the amplitude of the beatnote measured above threshold

consists out of 1% crosstalk, while 99% are originating from the beating of the pulse in the laser cavity. Hence, the measured beatnote amplitude of the AM state is hardly affected by the RF crosstalk. To clarify this, we have added the following sentence to the caption of figure 3:

“First three harmonics of the beatnote of the free-running 7 mm long QCL FM comb (red) compared to the actively mode-locked QCL (AM comb, blue) in logarithmic scale.”

We hope that with this letter, the revised manuscript and the supplementary information we could address all concerns of the reviewers.

Sincerely,

Benedikt Schwarz

REVIEWERS' COMMENTS

Reviewer #3 (Remarks to the Author):

I thank the authors in responding to my comments. The work is very good and of high quality

I apologise in being pedantic but the comment on THz QCLs should be corrected as longer lifetime than MIR QCL does not mean that active modelocking is easier, as suggested in the author's text. I would suggest "Note that in THz QCLs, where gain recovery times between 5ps and 50ps have been reported (please add in appropriate refs), actively mode-locked pulses have been reported in several works [21,22,23,24] with durations as short as 4 ps [25]".

I would suggest accepting the paper for Nature Comm after this minor modification

Vienna, 18.9.2020

Regarding: Revised manuscript NCOMMS-20-09590

Dear reviewer,

We are grateful for your valuable comments and your fast response. The manuscript was revised to address the reviewer's requests. The changes are given point by point in the following (reviewer comments in bold, answers in normal font, manuscript text in italic).

Reviewer #3 (Remarks to the Author):

I thank the authors in responding to my comments. The work is very good and of high quality

I apologise in being pedantic but the comment on THz QCLs should be corrected as longer lifetime than MIR QCL does not mean that active modelocking is easier, as suggested in the author's text. I would suggest "Note that in THz QCLs, where gain recovery times between 5ps and 50ps have been reported (please add in appropriate refs), actively mode-locked pulses have been reported in several works [21,22,23,24] with durations as short as 4 ps [25]".

I would suggest accepting the paper for Nature Comm after this minor modification

We thank the reviewer for the encouraging comment have changed this sentence according to this suggestion:

"Note that in THz QCLs, where gain recovery times between 5 ps and 50 ps have been reported²⁰, actively modelocked pulses were observed in several works^{21,22,23,24} with durations as short as 4 ps²⁵."

We hope that with this letter, the revised manuscript and the supplementary information we could address all comments of the editor and the reviewer.

Sincerely,
Benedikt Schwarz